

# Reducing uncertainty bounds of two-dimensional hydrodynamic model output by constraining model roughness

Punit Kumar Bhola, Jorge Leandro, Markus Disse

Chair of Hydrology and River Basin Management, Department of Civil, Geo and Environmental Engineering, Technical
University of Munich, Arcisstrasse 21 80333 Munich, Germany

*Correspondence to*: Punit K. Bhola (punit.bhola@tum.de)

**Abstract.** The consideration of uncertainties in flood risk assessment has received increasing attention over the last two decades. However, the assessment is not reported in practice due to the lack of best practices and too wide uncertainty bounds. We present a method to constrain the model roughness based on measured water levels and reduce the uncertainty bounds of a two-dimensional hydrodynamic model. Results show that the maximum uncertainty in roughness generated an uncertainty bound in the water level of 1.26 m (90% confidence interval) and by constraining roughness, the bounds can be reduced as much as 0.92 m.

## 1 Introduction

Uncertainties in flood risk assessment have received increasing attention from researchers over the last two decades. In Germany, flood risk management plans rely on hydrodynamic (HD) models to determine the impact of flooding for areas of potential flood risk (Thieken et al., 2016). Two-dimensional (2D) HD models are widely used to simulate flood hazards in the form of water depth, inundation extent and flow velocity (Disse et al., 2018). The hazard maps depict inundated areas for floods above certain exceedance levels, which leads to an improvement in flood risk assessment through increased spatial planning and urban development (Hagemeier-Klose, 2007).

Even though HD models are physically deterministic, they contain numerous uncertainties in model outputs (Bates et. al., 2014; Beven et al., 2018). Information about the type and magnitude of these uncertainties is crucial for decision making and to increase confidence in model predictions (Oubennaceur et al., 2018). Despite uncertainties, decision making in practice is based on first-hand data, expert judgement and/or a calibrated model output (Henonin et al., 2013; Uusitalo et al., 2015). Uncertainties associated with exceedance level scenarios are usually not quantified for at least five reasons: 1) most of the sources of uncertainty are not recognized (Bales and Wagner, 2009); 2) the data required to quantify uncertainty are seldom available (Werner et al., 2005); 3) high computational resources are required to perform an extensive uncertainty assessment; 4) the wide uncertainty bounds cannot be incorporated into the decision making process (Pappenberger and Beven, 2006); and 5) the uncertainty analysis is complex and is not considered for the final decision (Merwade et al., 2008).





The major sources of uncertainty in HD models can be categorized as model structure, model input, model parameters and the modeler (Matott et al., 2009; Schumann et al., 2011). The model structure, essentially either 1D, 2D or hybrid 1D-2D HD code, is generally selected based on the purpose and scale of the modelling (Musall et al., 2011; Bach et al., 2014). In addition, there is no general agreement on the best approach to consider model structure uncertainty; hence, it is often neglected

(Oubennaceur et al., 2018). In the case of hindcasting a flood event based on measured discharges or water levels as the input boundary conditions and a fine-resolution elevation, model parameters, mainly roughness, remain the main source of uncertainty in HD models, as is the case in this study.

The precise meaning of *roughness* changes based on a model's physical properties, such as grid resolution and time step (Bates et. al., 2014), and the term is denoted as Manning's roughness coefficient or simply Manning's n in most of HD models.

Various studies point out that HD models can be very sensitive to Manning's n, which implies a higher degree of uncertainty (Aronica et al., 1998; Pappenberger et al., 2005; Werner et al., 2005). The coefficient is either measured in the field or estimated from the relevant literature on the basis of land use types, but it has proven very difficult to demonstrate that such models can provide accurate predictions using only measured or estimated parameters (Hunter et al., 2007). In addition, Manning's n is not only related to bottom friction but also includes incorrect representation of turbulence losses, 3D effects and incorrect

geometry (profiles); therefore, it cannot be measured exactly.

In order to understand views on uncertainty analysis, it is important to look at the different modeler types. According to Pappenberger and Beven (2006), there are different modeler types: physically based modelers who believe that their models are physically accurate, and that the roughness must not be adjusted under any circumstances; the second modeler type believes that the roughness should be calibrated within a strictly known range (Wagener and Gupta, 2005); and the third modeler type

uses effective roughness beyond the accepted range (Pappenberger et al., 2005). The first modeler type would reject any calibration or uncertainty analysis; however, HD models make simplifying assumptions and do not consider all known processes that occur during a flood event (Romanowicz and Beven, 2003). Hence, models are subjected to a degree of structural errors that is typically compensated for by calibrating Manning's n (Bates et. al., 2014). However, effective roughness identified for one flood event might not hold true for another (Romanowicz and Beven, 2003), and a range of parameters

should be defined where equifinality can be observed.

Significant work has been done thus far in the quantification of HD model uncertainties and an overview of selected publications, including model roughness, is presented in Table 1. The major issue of wide uncertainty bounds raised by researchers and practitioners reflects in the table. It shows the maximum bounds reported in each publication and in some cases these bounds are more than 50% of the available water depth (Aronica et al., 1998; Hall et al., 2005; Werner et al., 2005; Jung

and Merwade, 2012). This is indeed an issue but not a reason to ignore uncertainties in predicting hazards. Moreover, decision makers must be made aware of potential risks associated with the possible outcomes of predictions, such as water levels and inundation extent (Pappenberger and Beven, 2006; Uusitalo et al., 2015).

The associated uncertainties can be constrained on measured data, if available, using a suitable goodness-of-fit or with the help of a sophisticated framework for assessment (Werner et al., 2005). Few researchers have used frameworks, such as *Generalized*



*Likelihood Uncertainty Estimation* (GLUE), the *Point Estimate Method* and *Global Sensitivity Analysis*, to reduce the bounds. These methods, although widely used in research, are not employed in operational practice, and a straightforward approach is needed to reduce the bounds.

This study investigates the use of measured water levels to reduce uncertainties bounds of HD model outputs. We begin with

the approach of the third modeler type and select extreme ranges of model roughness in literature and gradually shift to the approach of the second modeler type by reducing the uncertainty bounds based on the measured data. The main focus of this paper is to constrain literature-based ranges using measured water levels and to assess uncertainties in water levels. Uncertainty is quantified for the flood event of January 2011 in the city of Kulmbach, Germany.

## 2 Methods

To investigate the effect of measured data on constraining parameters, an ensemble of parameter sets was sampled using a prior distribution. In the HD model, distributed roughness values were used based on land use and a single value was used for each land use class. The model domain was spatially discretized based on the classification of land use and parameter sets were sampled using a prior. The choice of the distribution influences the outcome hence, it should be selected carefully. The 2D HD model was then run with each parameter set. The acceptance of each simulation was assessed by comparing the model

outputs and measured data. The measured data can be static or time-series water level measurements in the model domain and/or inundation extent gathered by field survey or post-event satellite images.

The performance of the simulations can be accessed using a suitable goodness-of-fit, such as Nash Sutcliffe efficiency, coefficient of determination, absolute error etc., based on the purpose of application and measured data available. A behaviour threshold was applied to divide simulations with acceptable performances from those with unacceptable performances.

Parameter sets that perform below the threshold were then selected at each location and an intersection at all the locations resulted in final number of accepted simulations (r) using equation 1

$$r = \bigcap_{i=1}^{n} P_i(GoF \leq e) \qquad\qquad (1)$$

Where n is the total number of observations, GoF is the goodness-of-fit used, e is the threshold and P is the array of models that satisfy the criteria of GoF below the threshold.

## 3 Materials

### 3.1 Study area and land use

The city of Kulmbach is located in the North-East of the federal state of Bavaria in Southern Germany. The city is categorized as a great district city inhabiting around 26,000 people and a population density of 280 inhabitants per $km^2$ in an area of 92.8 $km^2$. The city is crossed by the river White Main and Mühl canal, which is a diversion canal for flood protection. Schorgast

and Red Main are two main tributaries that meet the White Main upstream and downstream of the city respectively. In the





north, the small tributary Dobrach meets the White Main and from the south side, two storm water canals join the Mühl canal (see Figure 1). Main gauging stations upstream of the city are Ködnitz at White Main and Kauerndorf located at the river Schorgast.

The land use is shown in Fig. 1 and it generally consists of agricultural land (62%) that includes floodplains and grassland.

The water bodies make up 7% of the total model area and include rivers, canals and lakes. The urban area covers around 26% of the land and includes industrial and residential areas as well as transport infrastructures like roads and railway tracks, whereas forests form barely 5% of the total area. Fig. 2 shows images of the main channel and flood plain of the river White Main near site 1.

## 3.2 Measured discharges and water levels

Hydrological measurement data for the event was collected by the Bavarian Hydrological Services. Fig. 3 shows the discharge at main two gauges upstream of the city, Ködnitz and Kaurndorf, for the winter flood event of January 2011. It was one of the biggest in terms of its magnitude and corresponded to a discharge of the 100-year return period at gauge Kauerndorf and the 10-year return period at gauge Ködnitz. On 14th January, the peak discharge of 92.5 m³/s and 75.3 m³/s was recorded at gauge Kauerndorf and Ködnitz respectively.

Water levels at eight bridges during the winter flood of January 2011 were collected by the Water Management Authority in Hof, Germany in Kulmbach (see Fig. 1). The water levels were measured using a levelling instrument *Ni 2*. The instrument was used due to its availability and high accuracy; therefore, the measured levels were assumed to contain no error. Based on the locations, the sites are categorized in four groups: sites 1, 2, and 3 at the river White Main; site 4 at Dobrach canal in the north; site 5 at a side canal; and sites 6, 7, and 8 at Mühl canal.

**3.3 2D HD model**

HEC-RAS 2D was used as the 2D hydrodynamic model to quantify uncertainties in the inundation. The model uses an implicit finite difference solution algorithm to discretise time derivatives and hybrid approximations, combining finite differences and finite volumes to discretise spatial derivatives (Brunner, 2010). Table 2 shows the model properties and information of the cell size. We have used the diffusive wave model presented in previous work in Bhola et al. (2018a and 2018b).

Measured discharge hydrographs described in the previous section were used as the upstream boundary condition at river gauges Ködnitz, Kauerndorf and Unterzetlitz, and an energy slope value of 0.0096 at downstream boundary where the water flows out of the model domain. Along with the major rivers, canals were also represented as discharge hydrograph type.

High-quality digital elevation model for this study was provided by the Water Management Authority, Hof. In the elevation model, the terrain is determined by airborne laser scanning and airborne photogrammetry, whereas the river bed is mostly

recorded by terrestrial survey.



## 4 Results and discussion

For the study, we have performed 1000 simulations based on uniformly distributed parameter sets for five land use classes. Measured water levels at eight sites (see section 3.2) were used for the analysis of the model output. Absolute error between the simulated and measured water level is used as the goodness-of-fit to reach the objective.

### 4.1 Roughness range and distribution

The model parameter consists of roughness coefficient Manning's n for five land use classes. A simple model structure does not reflect the true distribution of the parameters in the basin. Hence, it is recommended to use extreme feasible upper and lower ranges for the parameters in literature (Aronica et al., 1998; Bhola et al., 2018b). In this study, ranges of Manning's n were set as: 0.015 – 0.15 for water bodies, which covers a range from very weedy reaches to rough asphalt; 0.025 – 0.110 for agriculture, short grass to medium-dense brush; 0.110 – 0.200 for forests, dense trees (Chow, 1959); 0.012 – 0.020 for transportation, firm soil to concrete; and 0.040 – 0.080 in urban areas, cotton fields to small boulders (Arcement and Schneider, 1989). Latin Hypercube sampling was used to generate 1000 parameter sets using the upper and lower ranges of Manning's n set as prior and HEC-RAS 2D model was simulated for each set.

### 4.2 Error tolerance

For the analyses, absolute error between the simulated and the measured water levels was calculated at eight sites. The simulations that produced an absolute error below a threshold at all the sites were selected. Fig. 4 shows that as we increase the threshold, the number of accepted simulations increases. To find one calibrated parameter set, the least value of tolerance can be set at 0.20 m that gives two simulation that results in least error at all site. Having said that, the calibrated roughness set will probably hold true only for the January 2011 event as discussed in the study (Romanowicz and Beven, 2003). In order to generalize the results to other events and collect enough samples to produce uncertainty bounds, the tolerance needs to be increased. In this study, we have used 1.5 m, 0.70 and 0.50 m as the tolerance at sites to evaluate the roughness sensitivity, which results in 1000, 339 and 143 selected simulations, respectively. Nevertheless, the tolerance can be changed depending on the requirements of the user. To summarize, three thresholds are used to evaluate the performance of the method in order to reduce the uncertainty bounds and are termed as follows

- Case I: Absolute error of 1.5 m resulting in 1000 simulations
- Case II: Absolute error of 0.7 m resulting in 339 simulations
- Case III: Absolute error of 0.5 m resulting in 143 simulations

### 4.3 Roughness sensitivity

Sensitivity of the model roughness was investigated, and it was observed that the sites were only sensitive to land use of water bodies and agriculture and no sensitivity was observed with respect to urban, transportation and forest. Table 3 presents the





coefficient of determination ($R^2$) between Manning's n for all the land uses and absolute error for case I. Site specific dependency in Manning's n and sites was observed for the cases in which the value of $R^2$ are found to be above *0.18* (in italic). The main reason for the sensitivity can be that the sites were mainly located on the bridges and are influenced either by water bodies or agriculture.

This was further restated in the scatter plots between the absolute error and Manning's n shown in Fig. 5. In the figure it can be observed that the cases II and II (with 339 and 143 accepted simulations) result in an absolute error of less than 0.70 m and 0.50 m at the sites respectively. The selected simulations were further used in refining the uncertainty bounds. Sites 1, 2 and 3 (White Main) show a pattern with agriculture (flood plain): as Manning's n increases, the error reduces until an optimal roughness is obtained and further increase in the roughness value results in increased error. Sites 6, 7 and 8, located at Mühl

canal, show similar sensitivity towards water bodies. In the case of sites 4 and 5, sensitivity is observed for both land use types. The sensitivity found here is also reflected in other studies, such as sensitivity to flood plains (agriculture) (Aronica et al., 1998) and main channel (water bodies) (Hall et al., 2005), and insensitivity to other land uses for flood events (Horritt and Bates, 2002; Werner et al., 2005).

## 4.4 Uncertainty of water levels

Table 4 presents 90% confidence interval absolute error bounds of simulated and measured water levels for three cases along with the measured available water depth. The average uncertainty bound was 0.87 m and after constraining with the measured data, it was reduced to 0.55 m for case II and further reduced to 0.38 in case III. The maximum bound of 1.26 m was observed at site 1, which was reduced to 0.59 and 0.34 m in case II and III respectively. Sites 7 and 8, located on Mühl canal, showed the least effect of 0.12 and 0.11 m reduction in the bounds respectively (case III). Fig. 6 presents a box plot of the difference

in the simulated and measured water levels. The pre-selected literature values of Manning's n tend to over-predict the water levels as the mean water level is well above zero at sites in case I. After constraining Manning's n, the mean drops considerably and is still above zero for all sites except 7 and 8 in both cases II and III. The figures also suggest that the simulations can both under- and over-predict the inundation, which might not be desired in some applications, such as early warning and evacuation planning. Furthermore, in situations where few sites are more sensitive/important than others a weighted goodness-of-fit can

also be realized. However, in this study we have focused on the overall uncertainties, both positive and negative, for a comprehensive assessment.

## 4.5 Constrained parameter set

The main objective of this study was to reduce the uncertainty bounds of the model output by constraining the prior set for the roughness. In this section, it is shown that the literature-based prior used for Manning's n can be reduced using measured water

levels. Fig. 7 presents the box plot of water bodies and agriculture roughness for three cases (1000, 339 and 143 accepted simulations). As stated in the previous section, no sensitivity was observed between the sites and other three land use types. Hence, the uncertainty bounds for other land use classes remain same after the analysis.



In the case of water bodies, Manning's n gradually concentrated in the range of 0.029 – 0.055 (25 – 75%, case III). The physical interpretation of the constrained coefficient ranges in main channels with stones to sluggish reaches (Chow, 1959). However, for agriculture the mean dropped considerably from case I to case II and remains consistent in case III. The 25 – 75% bound of the coefficient were 0.032 – 0.047 (case III) and can be interpreted as high grass to medium brush in the flood plains (Chow, 1959). This compares well to the results of Horritt and Bates (2002) in which they achieved an optimum in the range 0.03 – 0.05 for the main channel and 0.02 – 0.10 for the flood plain roughness of the 2D HD models.

Both the main channel and flood plains are homogenous in the model area and the presence of stones and high grass is observed in the field (see Fig. 2). It was discussed previously in the Introduction, that the second modeler type believes that Manning's n should be varied in a strictly known range based on field experiments. But these ranges can also be defined using a data-driven approach with the method presented. However, a detailed field experiments in the study area will be required to make a conclusive remark for a comparison between the field and evaluated coefficients. Furthermore, these ranges may vary for summer and winter events and various HD models can be build up depending on the season.

## 5 Conclusions

We have quantified the uncertainty associated with the model parameter for the flood event of January 2011 in the city of Kulmbach, Germany. Moreover, the study provides a comprehensive review of HD model uncertainty and explores the issue of high uncertainty bounds, which hinder users to analyse uncertainties. Extreme ranges of model roughness in the literature were selected and 1000 uniformly distributed models were run, which resulted in wide uncertainty bounds of up to 1.26 m (90% confidence interval). To reduce the bounds, measured water levels at eight sites were used and three cases were selected on the basis of absolute error threshold values of 1.5, 0.7 and 0.5 m, which resulted in 1000, 343 and 143 accepted simulations respectively. By constraining the roughness, the bounds were reduced to a maximum of 0.34 m. In addition, the model roughness was constrained, and the physical interpretation of the constrained roughness was discussed. The model roughness was spatially distributed based on five land uses and the model was sensitive only to water bodies and agriculture.

On an urban scale, the uncertainty assessment would substantially improve emergency responses by assessing potential consequences of flood events (Molinari et al., 2014), and disaster relief organisations, such as the Federal Agency for Technical Relief (THW), the German Red Cross, and the Bavarian Water Authorities, would indeed benefit from prioritising and coordinating evacuation planning. For advanced users such as decision-makers in water management authorities, the uncertainty assessment should further serve as a tool for enhanced risk assessment. In addition, by visualising inundation scenarios, improved flood mitigation and flood forecast planning strategies can be developed using a multi-model ensemble (Bhola et al., 2018d) and potential damage can be estimated for various quantiles.

Under-prediction of a simulated inundation is not desired in most case studies; therefore, the goodness-of-fit used in this study could be a critical issue. Future work should include other evaluation measures to constrain the parameter ranges. As the high-computational resources hinders a comprehensive uncertainty assessment of a full dynamic HD model, it is worth exploring





transferability of the evaluated uncertainty bounds of Manning's n of the simple model structure (diffusive wave) to a complex model structure. Furthermore, other sources of uncertainty, such as model input (hydrological model (Disse et al., 2018), discharge measurement error, or flood frequency estimations; and digital elevation map) and measured water level, which is assumed to have no error, should also be incorporated for a comprehensive assessment. The parameter ranges were constrained

based on a single event in this study; however, the values can be further validated using another flood event of higher magnitude. Land use in this study is divided into five classes; in future, further reclassification of land use, especially in urban areas, will help further reduce the bounds (Bhola et al., 2018c).

The inundation model should be extended to simulate urban pluvial flooding in future by including a 1D-2D sewer/overland flow coupled-model structure (Leandro et al., 2011). This will bring other sources of uncertainties as there are numerous

uncertain parameters associated with this model structure (Djordjević et al., 2014). With an ever-increasing computational performance and the introduction of cloud computing, the integration of more complex models will become feasible.

**Author contribution**

The study was conceptualized by PKB and MD, PKB conceptualized and completed the formal analysis of uncertainty analysis. PKB wrote the original draft was written by BGM and subsequently reviewed and edited by all co-authors. All authors

contributed to writing the paper.

**Competing interests**

The authors declare that they have no conflict of interest.

**Acknowledgements**

This research was funded by the German Federal Ministry of Education and Research (BMBF) with the grant number FKZ 13N13196. The authors would like to thank all contributing project partners, funding agencies, politicians, and stakeholders in different functions in Germany. A very special thanks to the Bavarian Water Authority and Bavarian Environment Agency in Hof for providing us the quality data to conduct the research. We would also like to thank the language centre of the Technical University of Munich for their consulting in English writing.

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

**Tables**

**Table 1. A summary of selected publications including the maximum uncertainty bound reported. GLUE, PEM, GSA and SD stands for *Generalized Likelihood Uncertainty Estimation*, *Point Estimate Method*, *Global Sensitivity Analyses*, and standard deviation respectively.**

| Model dimension | HD Model | Identified sources | Method | Sample size | Max bound | Literature |
|---|---|---|---|---|---|---|
| 1D | HEC-RAS | Manning's n | GLUE | 10000 | ~ | Pappenberger et al. (2005) |
| 1D | HEC-RAS | Flow Topography Manning's n | GLUE | 5000 | ~2.5 m (95%) in 8 m | Jung and Merwade (2012) |
| 1D-2D | SOBEK | Topography Manning's n | GLUE | | 1.64 m (90%) in 1.51 m | Werner et al. (2005) |
| 2D | | Manning's n | GLUE | 1000 | ~7m (90%) in 10.5 m | Aronica et al. (1998) |
| 2D | H2D2 | Flow Topography Manning's n | PEM | 108 | 0.27 m SD in 12.06 m | Oubennaceur et al. (2018) |
| 2D | Lisflood-FP | Flow Topography Manning's n Channel width | GSA | 1792 | 6 m SD in 11 m | Hall et al. (2005) |

**Table 2. 2D hydrodynamic model properties.**



| Data | Value |
|---|---|
| Model area | 11.5 km$^2$ |
| Total number of cells | 430,485 |
| Δt | 20 s |
| Minimum cell area | 6.8 m$^2$ |
| Maximum cell area | 59.8 m$^2$ |
| Average cell area | 24.8 m$^2$ |

**Table 3. Coefficient of determination ($R^2$) between Manning's n and absolute error for case I.**

| Site | Coefficient of determination [-] | | | | |
|---|---|---|---|---|---|
| | Water bodies | Agriculture | Forest | Transportation | Urban |
| 1 | 0.04 | *0.89* | 0.00 | 0.00 | 0.00 |
| 2 | 0.05 | *0.85* | 0.00 | 0.00 | 0.00 |
| 3 | *0.18* | 0.69 | 0.00 | 0.00 | 0.00 |
| 4 | *0.34* | 0.54 | 0.00 | 0.00 | 0.01 |
| 5 | *0.45* | 0.37 | 0.00 | 0.00 | 0.00 |
| 6 | *0.97* | 0.00 | 0.00 | 0.00 | 0.00 |
| 7 | *0.23* | 0.18 | 0.00 | 0.00 | 0.00 |
| 8 | *0.19* | 0.22 | 0.00 | 0.00 | 0.00 |

**Table 4. 90% confidence interval absolute error bounds (in m) for three cases along with measured water depth (in m) at eight sites for the January 2011 event.**

| Site | Measured water depth[1] | 90% absolute error bounds | | |
|---|---|---|---|---|
| | | Case I | Case II | Case III |
| 1 | 2.78 | 1.26 | 0.59 | 0.34 |
| 2 | 2.90 | 1.04 | 0.55 | 0.34 |
| 3 | 2.93 | 1.01 | 0.59 | 0.36 |
| 4 | 1.43 | 0.97 | 0.64 | 0.46 |
| 5 | 1.75 | 0.78 | 0.46 | 0.32 |
| 6 | 0.89 | 0.85 | 0.65 | 0.43 |
| 7 | 2.31 | 0.52 | 0.46 | 0.40 |
| 8 | 2.36 | 0.51 | 0.46 | 0.40 |



**Figures**

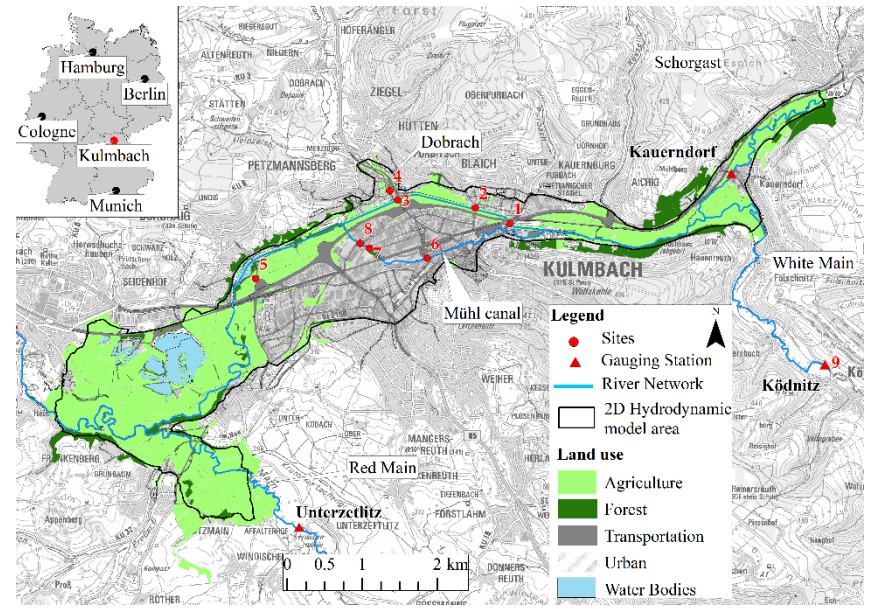

**Figure 1: Land use of the city of Kulmbach.**

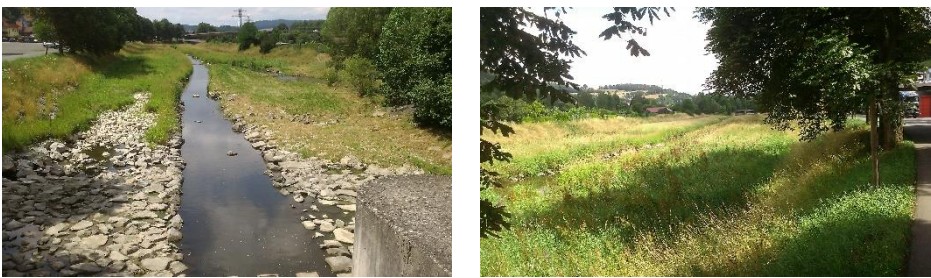

**Figure 2: Main channel and flood plain of the river White Main near site 1 (image taken on 23.07.2015).**





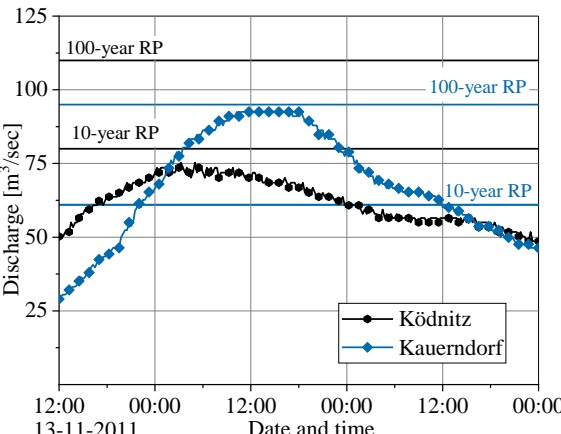

**Figure 3: Discharge hydrographs at gauging stations upstream of the city, Ködnitz and Kauerndorf. RP stands for return period. Data source: Bavarian Hydrological Service (www.gkd.bayern.de).**

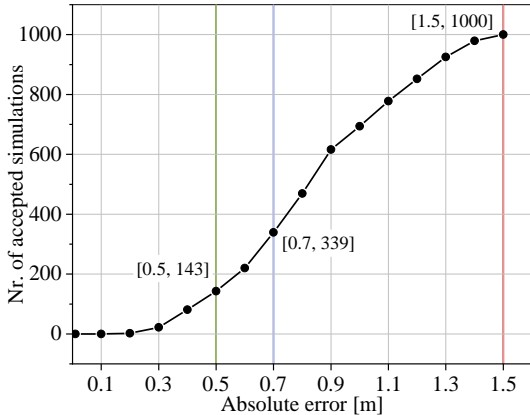

**Figure 4: Accepted number of simulations vs. absolute error.**

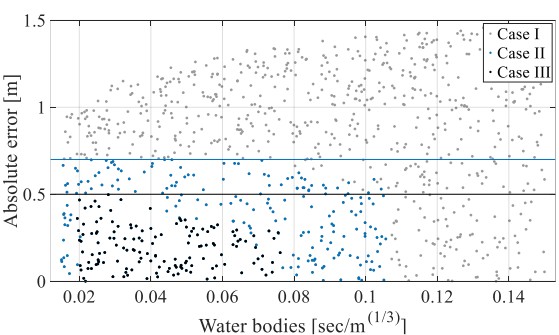

**(a) Site 1: Water bodies**

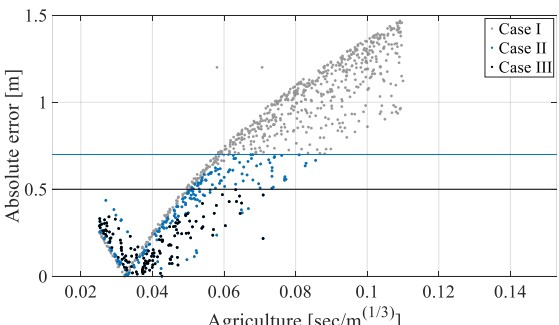

**(b) Site 1: Agriculture**




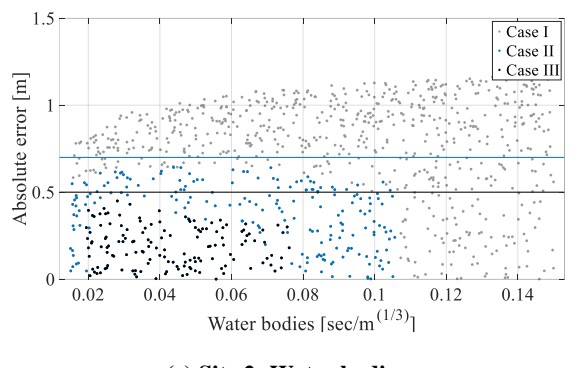

**(c) Site 2: Water bodies**

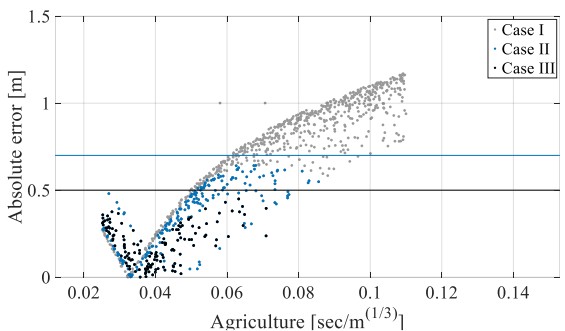

**(d) Site 2: Agriculture**

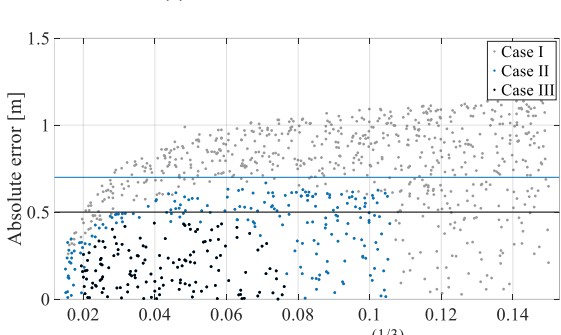

**(e) Site 3: Water bodies**

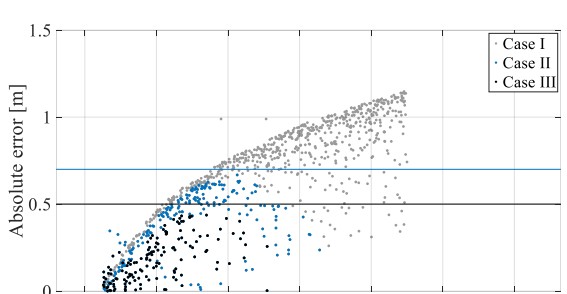

**(f) Site 3: Agriculture**

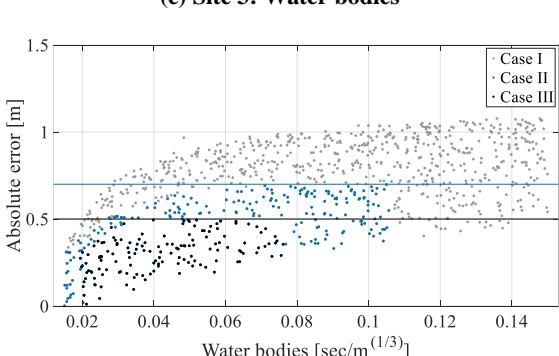

**(g) Site 4: Water bodies**

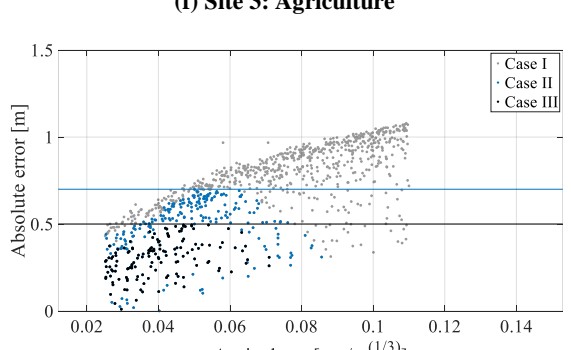

**(h) Site 4: Agriculture**



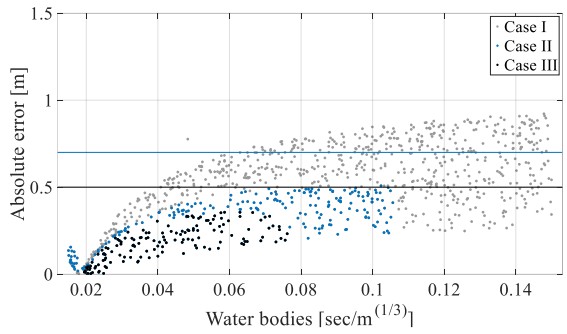

**(i) Site 5: Water bodies**

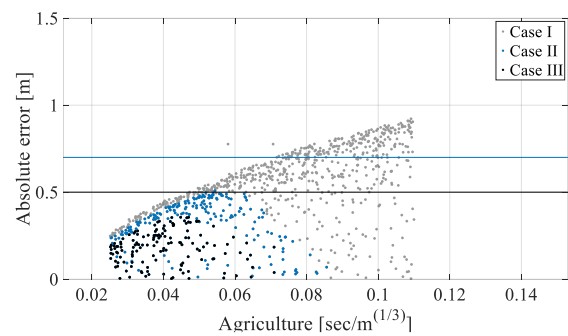

**(j) Site 5: Agriculture**

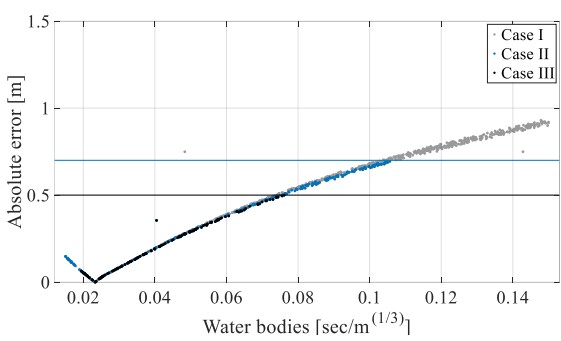

**(k) Site 6: Water bodies**

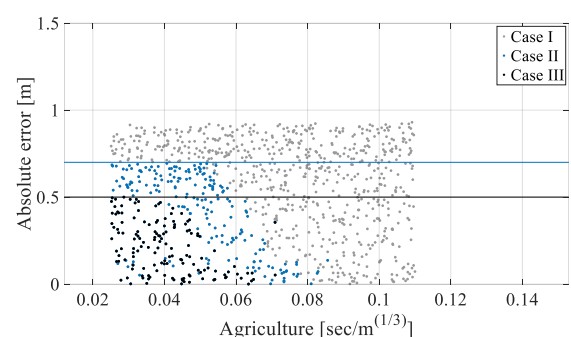

**(l) Site 6: Agriculture**

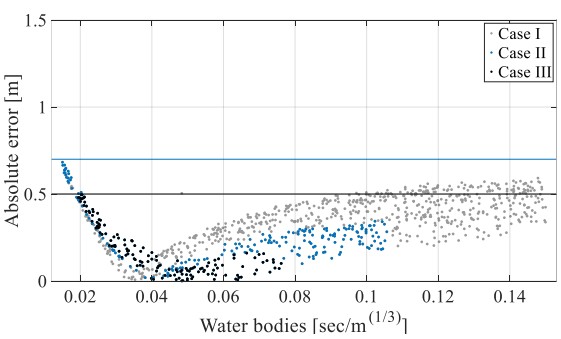

**(m) Site 7: Water bodies**

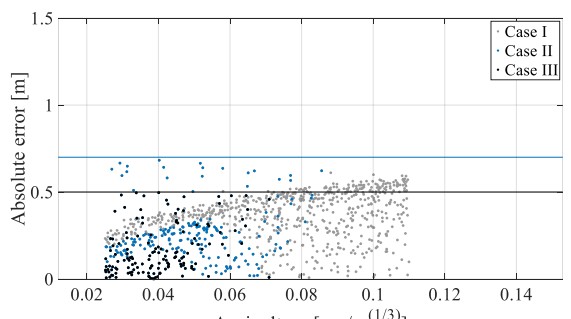

**(n) Site 7: Agriculture**





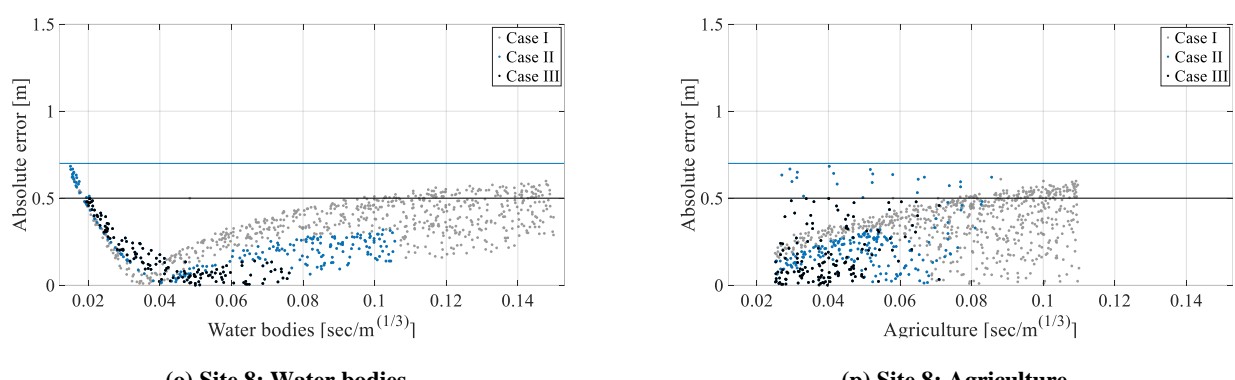

**(o) Site 8: Water bodies**       **(p) Site 8: Agriculture**

**Figure 5:** Scatter plot of the absolute error of 1000 simulation in relation to water bodies and agriculture. Three cases I, II and III shows accepted simulations based on threshold values of 1.5, 0.7 and 0.5 m respectively.

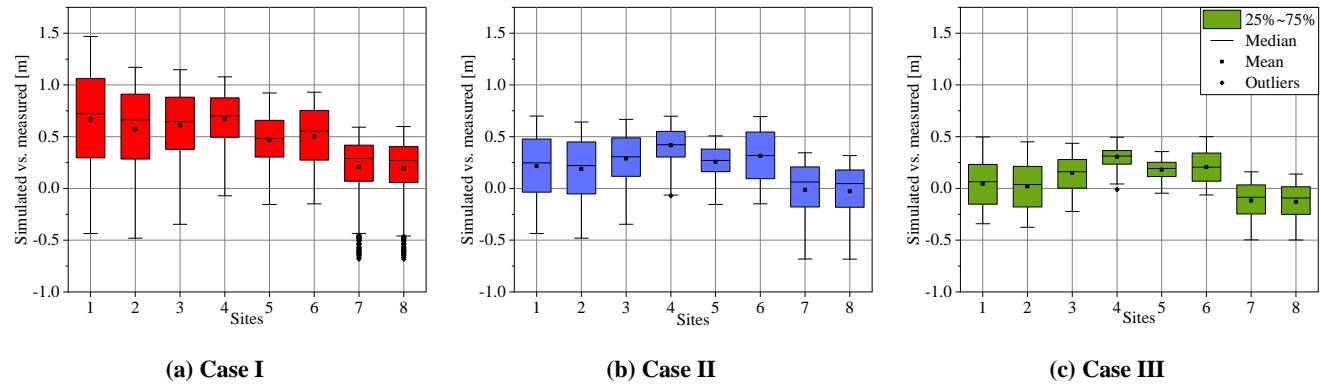

**(a) Case I**       **(b) Case II**       **(c) Case III**

**Figure 6:** Error in simulated vs. measured water levels for a) Case I, b) Case II, and c) Case III.

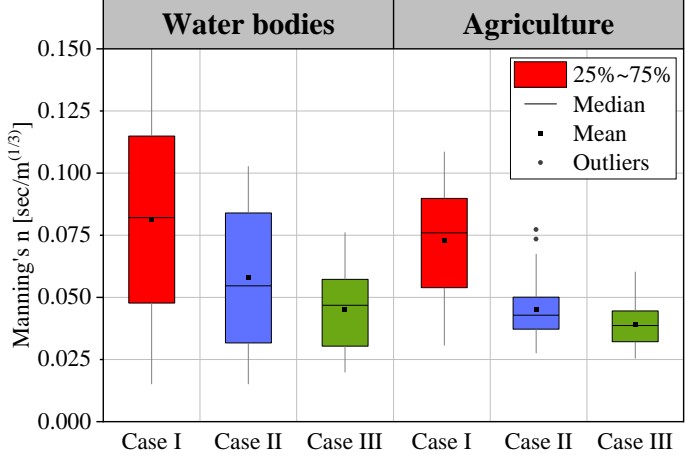

**Figure 7:** Box plot of Manning's n of water bodies and agriculture for three cases.