# Peer review of "Reducing uncertainties in flood inundation outputs of a twodimensional hydrodynamic model by constraining roughness"

_Natural Hazards and Earth System Sciences, 2018_

## Referee Comment (RC1) · Anonymous Referee #1 · 12 Mar 2019

**General Comments ------**

Thanks for this interesting and informative paper. In general the paper was easy to read, well-structured and complete. The topic of the paper fits to the scope of Natural Hazards and Earth System Sciences in this case to flooding and inundation. The content is relevant to the scientific community and gives substantial contribution to the knowledge in this domain.

The paper is in his core a case study for a specific flood event in Germany. I recommend to specify the natural hazards and target of the 2D modelling (flood-ing/inundation) in the title. A generalization of the method to other case studies or

other application scenarios would be beneficial for the reader. This could be done in a minor or a major revision. However, the conclusions of the work done should be described and highlighted beyond the model case study in Germany as added value for the reader of the paper. The question, in which way the paper can help other modellers for other case studies, should be answered explicitly.

Detailed Comments -----

Page 2 line 11 The coefficient is either measured in the field -> How did you measure the Manning coefficient in the field ? I think the coefficient is not measured but derived from measurements. see same page line 15 -> "therefore, it cannot be measured exactly"

Introduction The spatial distribution of the roughness (e.g. in the river bed, in flood plains and in areas with inundation), is missing in the introduction, maybe you can add 1,2 sentences to clarify this problem/challenge.

The last sentence about the case study in Kulmbach is too short, reading up to here, it is not clear how the flood is triggered (heavy rain -> flash flood in the city, river flood wave inundation, ...) maybe you can add some key words / numbers to classify the case study area in more sentence to characterize the type of flood case study. 3.1 will gives the details later

Page 4 line 15 Please add the term "site" in the sentence, as the 8 bridges are shown as site 1-8 in Figure 1. line 16 Please add a reference to the levelling instrument Ni2 and add some numbers to "high accuracy".

line 26 Unterzetlitz is not mentioned in chapter 3.2 as "previous section", please add a sentence in 3.2 as only Kodnitz and Kaurndorf are mentioned and only these two hydrographs are shown. Maybe the discharge of Red Main / Unterzetlitz is not critical for the flooding area, but if you mention it in 3.3 the information should be complete

line 28 What is "high-quality" DEM, please mention the resolution e.g. in m and how

this is represented and combined in the numerical mesh (terrain, river bed)?

Chapter 3.2 the time steps given in table 2, assuming it is a global time step. I'm missing in the paper the simulation period (2,5 days ?) maybe one sentence about the required time for one simulation on a specified hardware would be also helpful. Not everyone is reading your two papers 2018a and 2018b before reading this paper, as only 2018b is available at the moment

Page 5 line 2 Why did you performed 1000 simulations, not 500, not 2000, please add a sentence to validate the number of performed simulations

line 6 What is a "simple" model ? Is there any model reflecting the "true" distribution of the parameters ? what is the "basin" ? Do you mean model region/area ? line 11 cotton fields and small boulders are confusing here

Page 6 line 6 I instead of II (2\*II)

line 3 ff This sentence is not clear for me: The sites are located at bridges in the water bodies, this is clear. But the water level is less depending on the landuse at the location more on the upstream flow situation, which type of landuse is there and how much area is flooded (besides water bodies) at upstream. In Figure 2 is shown that the main upstream landuse is water body and agriculture and only for extended inundation small areas of forest, urban region and transportation -> as there is no Figure with a DEM or with a typical flow situation (flood map), the impact is not clear except for site 1. I propose to add at least one flood map to argument for this sensitivity. As all this is important for interpreting the Figure 5 (a) - (p), maybe a more detailed, structured description of the landuse impact to site 1-8 might be helpful. If finally only two roughness coefficients are sensitive (as first result of the analysis), why the three other parameters has not been eliminated for a 2nd step to rerun the 1000 simulations (with two parameters only, maybe less simulations are needed) and to concentrate on the changes of these two parameters in the parameter sets?

Page 7 The conclusion is more a summary and a outlook, but no conclusion. A generalization of the finding is missing, in which way the reader can benefit from this study ? Are there recommendations to do a similar approach in similar case or a general guideline to reduce the uncertainty bounds for flood modelling

What is the consequence for the flood modelling for the city of Kulmbach ? Maybe this question can be discussed in Chapter 4. I can follow the description in Chapter 4, but I missed the consequences for the modelling tasks.

Page 8 line 13-15 the short-cuts for the authors should be not used, who is BGM ?

Figure 1 Why did you not use the full width of the page? the figure size could be increased

Figure 2 1st image is fine, but difficult to discover the river bed without zooming in

---

## Referee Comment (RC2) · Guy J.-P. Schumann (Referee) · 13 Mar 2019

This paper describes a case study of constraining the model roughness parameter as a means to reduce the overall uncertainty in 2D inundation models.

In general, the paper is well written and, as so many papers around this topic that now start to become quite dated, is an interesting read and debates a very important topic: quite straightforward uncertainty reduction methods are available and should be used and applied much more in practice. Although, this argument was made a lot quite some years ago, I kind of welcome this paper, as it refreshes this important point.

[Figure]

Here are some points that I feel need to be addressed before publication:

In my mind Keith Beven and Florian Pappenberger wrote two of the best papers on this topic, both in 2006 so 13 (or more) years ago, namely: Beven: https://www.sciencedirect.com/science/article/pii/S002216940500332X Pappenberger: https://agupubs.onlinelibrary.wiley.com/doi/10.1029/2005WR004820

While the latter is cited by the authors, the former is not I believe and I think it should because I think it would be very useful in this presented study if the authors put their work in context of those two papers and build a justification around them to state why their presented case study is needed and what makes it different to existing literature, which, although now dated, is substantially large, especially the the 10 years 1998-2009.

Without such a "putting in context", this paper only really refreshes this very well known problem. It is my opinion, that with such a justification, the paper could be published subject to "minor/moderate" revisions but without it, I think it is unclear what new message is presented here.

Also, the authors need to clarify why they did not consider other sources of uncertainty in their model, such as discharge or downstream boundary condition or indeed topography? Why only roughness? Also, they should explain why they decided to do 1000 simulations and how this number was decided?

---

## Author Response (AR1)

Referee 1:

We sincerely appreciate the constructive feedback from the reviewer in improving the quality of the manuscript. We have addressed all the review comments in the revised manuscript.

Anonymous Referee:

Thanks for this interesting and informative paper. In general the paper was easy to read, well-structured and complete. The topic of the paper fits to the scope of Natural Hazards and Earth System Sciences in this case to flooding and inundation. The content is relevant to the scientific community and gives substantial contribution to the knowledge in this domain.

The paper is in his core a case study for a specific flood event in Germany. I recommend to specify the natural hazards and target of the 2D modelling (flooding/ inundation) in the title. A generalization of the method to other case studies or other application scenarios would be beneficial for the reader. This could be done in a minor or a major revision. However, the conclusions of the work done should be described and highlighted beyond the model case study in Germany as added value for the reader of the paper. The question, in which way the paper can help other modellers for other case studies, should be answered explicitly.

Authors:

The title has been updated to specify the hazard. The new title following the recommendation is

"Reducing uncertainties in flood inundation outputs of a two-dimensional hydrodynamic model by constraining roughness"

The conclusion section has been revised in the updated manuscript in order to generalization results to other case studies. In addition, more discussion is added as to how this paper can help other modellers.

The followings are our point-by-point responses to the reviewer's quotes

Anonymous Referee:

1.      Page 2 line 11 The coefficient is either measured in the field -> How did you measure the Manning coefficient in the field ? I think the coefficient is not measured but derived from measurements. see same page line 15 -> "therefore, it cannot be measured

exactly"

Authors:

The line has been corrected to "derived from measurements in the field". Page 2: Line 11-12

"and the term is denoted as Manning's roughness coefficient or simply Manning's n in most of HD models"

Anonymous Referee:

2.      Introduction- The spatial distribution of the roughness (e.g. in the river bed, in flood

plains and in areas with inundation), is missing in the introduction, maybe you can add

1,2 sentences to clarify this problem/challenge.

Authors:

To clarify the problem of the spatial distribution, these sentences have been added in introduction. Page 2:Line 15-17.

"The spatial distribution of the Manning's n in floodplains is challenging and depend on many factors, such as vegetation type, soil surface and imperviousness (Sellin et al., 2013). Traditionally, this coefficient can be best estimated based on lookup tables of land use types (Werner et al., 2005b)."

Anonymous Referee:

3.      The last sentence about the case study in Kulmbach is too short, reading up to here,

it is not clear how the flood is triggered (heavy rain -> flash flood in the city, river flood

wave inundation, ...) maybe you can add some key words / numbers to classify the

case study area in more sentence to characterize the type of flood case study. 3.1 will

gives the details later

Authors:

We have provided more details of the flood event. It was induced by a combination of intense rainfall and snowmelt in the catchment. Page 4: Line 17-23

"Intense rainfall and snow melting in the Fichtel mountains caused floods in several rivers of Upper Franconia. On 14th January, the maximum discharge of 92.5 m³/s was recorded at gauge Kauerndorf and 75.3 m³/s at gauge Ködnitz. It was one of the biggest in terms of its magnitude and corresponded to a discharge of the 100-year return period at gauge Kauerndorf and the 10-year return period at gauge Ködnitz. Agricultural land and traffic

routes were flooded, but no serious damage was reported. In Kulmbach, a dyke in the region of Burghaig was about to collapse due to the large volume of water. The Water Management Authority opened the weir in Kulmbach which saved potential damages (Hof, 2011)."

Anonymous Referee:

4.      Page 4 line 15 Please add the term "site" in the sentence, as the 8 bridges are shown

as site 1-8 in Figure 1. line 16 Please add a reference to the levelling instrument Ni2

and add some numbers to "high accuracy".

Authors:

The measurements were taken by the experts in the Water Management Authority in Hof, Germany during the flood event. It is believed that they are accurate, however, we are not able to provide any numbers regarding the accuracy. The instrument has its own uncertainties (details in Faig and Kahmen, 2012) but it will be very subjective to add in this paper and not the scope of this paper.

The term "bridges" has been corrected and reference to the instrument has been provided. Page 4: Line 24-25.

"Water levels at eight sites during the winter flood of January 2011 were collected by the Water Management Authority in Hof, Germany in Kulmbach (see Fig. 1a). The water levels were measured using a levelling instrument Ni 2 (Faig & Kahmen 2012)."

Anonymous Referee:

5.      line 26 Unterzetlitz is not mentioned in chapter 3.2 as "previous section", please add

a sentence in 3.2 as only Kodnitz and Kaurndorf are mentioned and only these two

hydrographs are shown. Maybe the discharge of Red Main / Unterzetlitz is not critical

for the flooding area, but if you mention it in 3.3 the information should be complete

Authors:

Based on the recommendation, we have omitted Unterzetlitz from the manuscript. The river Red Main (gauging station Unterzetlitz) is downstream of the city and the eight sites assessed in the study. Hence, we believe that presenting discharge data for that particular gauge is irrelevant for this study. Nevertheless, all the discharge data is open and downloadable from the data source provided in Fig. 3: Bavarian Hydrological Service (www.gkd.bayern.de).

Anonymous Referee:

6.      line 28 What is "high-quality" DEM, please mention the resolution e.g. in m and how

this is represented and combined in the numerical mesh (terrain, river bed) ?

Authors:

The description has been updated in Page 5: Line 7-12.

"Digital elevation model for this study was provided by the Water Management Authority, Hof and presented in Fig. 1b. In the provided elevation model, the terrain is determined by airborne laser scanning and airborne photogrammetry with a high-resolution of 1 meter, whereas the river bed was mostly recorded by the terrestrial survey. The combined elevation data were used to generate a triangulated irregular network (TIN) of the topography, which was then resampled to an irregular mesh of the 2D HD model. Special attention was given in resampling in order to preserve important features, such as rivers, dykes, buildings and roads."

Anonymous Referee:

7.      Chapter 3.2 the time steps given in table 2, assuming it is a global time step. I'm missing in the paper the simulation period (2,5 days ?) maybe one sentence about the required time for one simulation on a specified hardware would be also helpful. Not everyone is reading your two papers 2018a and 2018b before reading this paper, as only 2018b is available at the moment.

Authors:

More details regarding the simulation time is provided in revised version and also in Table 2 for readers .Page 5: Line 16-20

"The HD models were simulated starting at 13.01.2011 00:00 to 14.01.2011 18:00, which requires approximately five hours to simulate an event of 42 hours on an eight-core, Intel® Core™ 2 Duo CPU T7700 @ 2.40 cloud computer with 64 GB RAM. Eight cloud computers using the LRZ Compute Cloud, provided by the Leibniz Supercomputing Centre of the Bavarian Academy of Sciences and Humanities, were used to complete 1000 simulation in two weeks."

Anonymous Referee:

8.      Page 5 line 2 Why did you performed 1000 simulations, not 500, not 2000, please add a sentence to validate the number of performed simulations

Authors:

A justification is given in the revised manuscript behind 1000 simulations on Page 5: Line 14-17.

"For the study, we have performed 1000 simulations based on uniformly distributed parameter sets for five land use classes. The sample size does contain enough samples of different behavioural models and the estimate was based on the recommendation in the literature (Aronica et al., 1998; Romanowicz and Beven, 2003) as well as the computational resources available."

Anonymous Referee:

9.       line 6 What is a "simple" model ? Is there any model reflecting the "true" distribution

of the parameters ? what is the "basin" ? Do you mean model region/area ? line 11

cotton fields and small boulders are confusing here

Authors:

The sentence has been updated in Page 5: Line 24-26

"The model parameter consists of roughness coefficient Manning's n for five land use classes. A simple model structure, such as diffusive wave approximation, does not represent the accurate values of roughness as this parameter is scale-dependent effective values that compensate for varying conceptual errors in the model (Néelz et al., 2009)."

In addition, cotton fields and small boulders have been updated to parks to gravels Page 5: Line 30

"0.040 – 0.080, parks to gravels in urban areas (Arcement and Schneider, 1989)."

Anonymous Referee:

10.      Page 6 line 6 I instead of II (2*II)

Authors: Thank you for pointing it out, it has been corrected to cases II and III

Anonymous Referee:

11.      line 3 ff This sentence is not clear for me: The sites are located at bridges in the

water bodies, this is clear. But the water level is less depending on the landuse at

the location more on the upstream flow situation, which type of landuse is there and

how much area is flooded (besides water bodies) at upstream. In Figure 2 is shown

that the main upstream landuse is water body and agriculture and only for extended

inundation small areas of forest, urban region and transportation -> as there is no

Figure with a DEM or with a typical flow situation (flood map), the impact is not clear

except for site 1. I propose to add at least one flood map to argument for this sensitivity.

As all this is important for interpreting the Figure 5 (a) - (p), maybe a more detailed,

structured description of the landuse impact to site 1-8 might be helpful. If finally only

two roughness coefficients are sensitive (as first result of the analysis), why the three

other parameters has not been eliminated for a 2nd step to rerun the 1000 simulations

(with two parameters only, maybe less simulations are needed) and to concentrate on

the changes of these two parameters in the parameter sets?

Authors:

We have added both DTM in Fig. 1b and a flood inundation map in Fig. 5 of the study area for the flood event of January 2011.

We appreciate the reviewers' suggestion to concentrate on the changes of these two parameters, however, two roughness were found out to be sensitive as an outcome of the methodology and the idea or focus of this study was not to perform an iterative analysis. We believe that in other study areas using another model structure, might show sensitivity towards other land uses as well. In addition, we also believe that the sensitivity also depends on the land use type of the site. A conclusive remark can only be made if the measured sites are evenly distributed in the model domain in all the land uses

Furthermore, a description of the land use impact has been provided in Page 6: Line 22-27.

"The main reason for the lack of sensitivity can be explained by the location of the sites since they were mainly located next to bridges upstream from water bodies or agriculture land uses. Nonetheless, there are other influencing factors, such as the inundation area, velocity, and topography that could also play a role (Werner et al., 2005b). Fig. 5 shows the maximum flood inundation map for the January 2011 flood event simulated using the optimal model parameters, which were obtained by the least absolute error of 0.20 m. The inundation upstream to the sites is mainly constrained in the water bodies and agricultural land uses, which explains the impact on sensitivity of water levels to these two land uses."

Anonymous Referee:

12.      Page 7 The conclusion is more a summary and a outlook, but no conclusion. A generalization of the finding is missing, in which way the reader can benefit from this study

? Are there recommendations to do a similar approach in similar case or a general

guideline to reduce the uncertainty bounds for flood modelling

Authors:

We thank the reviewer for pointing it out and we have updated to improve the conclusion section. This part has been added to the generalization of the finding. Page 8: Line 17-22.

"The method is easy to incorporate into other study areas, provided that there are measured water levels available. The uncertainty analysis presented in this study allows a better understanding of the model roughness variability in HD models. The ranges researched for Manning's n in this study can represent a good starting point (prior distribution) for other studies. Our study has shown that there are significant uncertainties in HD model roughness and should be considered in decision-making. In addition, the study highlights the importance of field surveys for reducing the uncertainty in flood inundation outputs."

Anonymous Referee:

What is the consequence for the flood modelling for the city of Kulmbach ? Maybe this question can be discussed in Chapter 4. I can follow the description in Chapter 4, but I missed the consequences for the modelling tasks.

Authors:

The consequence for the flood modelling is highlighted in chapter 4.4. Page 7: Line 7-8

"The impact of reducing the uncertainty is clear in the simulated flood inundation for the city of Kulmbach".

Anonymous Referee:

13.     Page 8 line 13-15 the short-cuts for the authors should be not used, who is BGM ?

Authors:

BGM was there by mistake, thank you for pointing it out. The short cut of authors are as per the recommendation of author contributions guidelines and kept the same.

Anonymous Referee:

14.     Figure 1 Why did you not use the full width of the page? the figure size could be increased

Authors:

The figure has been increased to full page width. Nevertheless, all the figures will be provided in original form and high-resolution of 300dpi to the journal to ensure high quality.

Anonymous Referee:

15.     Figure 2 1st image is fine, but difficult to discover the river bed without zooming in

Authors:

We have replaced Figure 2b, where the river bed is visible, however, there is still some vegetation cover.

Referee 2

Guy J.-P. Schumann:

This paper describes a case study of constraining the model roughness parameter as a means to reduce the overall uncertainty in 2D inundation models.

In general, the paper is well written and, as so many papers around this topic that now

start to become quite dated, is an interesting read and debates a very important topic:

quite straightforward uncertainty reduction methods are available and should be used

and applied much more in practice. Although, this argument was made a lot quite some

years ago, I kind of welcome this paper, as it refreshes this important point.

Authors:

We sincerely appreciate the detailed and positive feedback from the reviewer. We also second the opinion of the reviewer that the argument of uncertainty reduction is outdated, especially in operational-use. We hope that we have addressed all the comments satisfactorily in the revised manuscript, which improves the quality of this paper.

The followings are our point-by-point responses to the reviewer's quotes

Guy J.-P. Schumann:

Here are some points that I feel need to be addressed before publication:

In my mind Keith Beven and Florian Pappenberger wrote two of the best

papers on this topic, both in 2006 so 13 (or more) years ago, namely:

Beven:                    https://www.sciencedirect.com/science/article/pii/S002216940500332X
Pappenberger:

https://agupubs.onlinelibrary.wiley.com/doi/10.1029/2005WR004820

While the latter is cited by the authors, the former is not I believe and I think it should

because I think it would be very useful in this presented study if the authors put their

work in context of those two papers and build a justification around them to state why

their presented case study is needed and what makes it different to existing literature,

which, although now dated, is substantially large, especially the the 10 years 1998-

2009.

Without such a "putting in context", this paper only really refreshes this very well known

problem. It is my opinion, that with such a justification, the paper could be published

subject to "minor/moderate" revisions but without it, I think it is unclear what new message

is presented here.

Authors:

We thank the reviewer for this suggestion, The reference Beven (2006) has been added in the review in the revised manuscript. Page 2: Line 24-29.

"However, effective roughness identified for one flood event might not hold true for another (Romanowicz and Beven, 2003), and a range of parameters should be defined where equifinality can be observed. Beven (2006) argued that the prior selected for the range of parameters should potentially cover all the accepted or behavioural models (modeller types 2 or 3). In HD models, selecting such a prior distribution for model parameter introduces the issue of too wide bounds."

In addition, the novelty or the research gap has been clearly addressed in the revised version. Page 3: Line 6-8.

"These methods, although widely used in research, are not employed in operational practice, and a straightforward approach is needed to reduce the bounds. Furthermore, there is a need to ensure efficiency in searching model parameter spaces for behavioural models (Beven, 2006)."

Guy J.-P. Schumann:

Also, the authors need to clarify why they did not consider other sources of uncertainty

in their model, such as discharge or downstream boundary condition or indeed topography?

Why only roughness? Also, they should explain why they decided to do 1000

simulations and how this number was decided?

Authors:

A justification is given in the revised manuscript behind 1000 simulations on Page 5: Line 14-17.

"For the study, we have performed 1000 simulations based on uniformly distributed parameter sets for five land use classes. The sample size does contain enough samples of different behavioural models and the estimate was based on the recommendation in the literature (Aronica et al., 1998; Romanowicz and Beven, 2003) as well as the computational resources available."

In addition, more information is provided as to why other sources were not considered in this paper, Page 2: Line 5-7

"In the case of hindcasting a flood event based on measured discharges or water levels as the input boundary conditions and a fine-resolution elevation, roughness remains the main source of uncertainty in HD models; hence we focus this study on roughness uncertainty."

**Reducing uncertainties in flood inundation outputs of a two-dimensional hydrodynamic model by constraining roughness**

Punit Kumar Bhola, Jorge Leandro, Markus Disse

[revised manuscript text omitted]

15 effects and incorrect geometry (profiles); therefore, it cannot be measured exactly. The spatial distribution of the Manning's n in floodplains is challenging and depend on many factors, such as vegetation type, soil surface and imperviousness (Sellin et al., 2013). Traditionally, this coefficient can be best estimated based on lookup tables of land use types (Werner et al., 2005b).

In order to understand views on uncertainty analysis, it is important to look at the different modeller types. According to Pappenberger and Beven (2006), there are different modeller types: physically based modellers who believe that their models

20 are physically accurate and that the roughness must not be adjusted under any circumstances; the second modeller type believes that the roughness should be calibrated within a strictly known range (Wagener and Gupta, 2005); and the third modeller type uses effective roughness beyond the accepted range (Pappenberger et al., 2005). The first modeller type would reject any calibration or uncertainty analysis; however, HD models make simplifying assumptions and do not consider all known processes that occur during a flood event (Romanowicz and Beven, 2003). Hence, models are subjected to a degree of structural

25 errors that are typically compensated for by calibrating Manning's n (Bates et. al., 2014). However, effective roughness identified for one flood event might not hold true for another (Romanowicz and Beven, 2003), and a range of parameters should be defined where equifinality can be observed. Beven (2006) argued that the prior selected for the range of parameters should potentially cover all the accepted or behavioural models (modeller types 2 or 3). In HD models, selecting such a prior distribution for model parameter introduces the issue of too wide bounds.

[revised manuscript text omitted]

15 **3.2 Measured discharges and water levels**

Hydrological measurement data for the winter flood event of January 2011 was collected by the Bavarian Hydrological Services. Fig. 3 shows the discharge at the main two gauges upstream of the city, Ködnitz and Kaurndorf. Intense rainfall and snow melting in the Fichtel mountains caused floods in several rivers of Upper Franconia. On 14th January, the maximum discharge of 92.5 m³/s was recorded at gauge Kauerndorf and 75.3 m³/s at gauge Ködnitz. It was one of the biggest in terms

20 of its magnitude and corresponded to a discharge of the 100-year return period at gauge Kauerndorf and the 10-year return period at gauge Ködnitz. Agricultural land and traffic routes were flooded, but no serious damage was reported. In Kulmbach, a dyke in the region of Burghaig was about to collapse due to the large volume of water. The Water Management Authority opened the weir in Kulmbach which saved potential damages (Hof, 2011).

Water levels at eight sites during the winter flood of January 2011 were collected by the Water Management Authority in Hof, 25 Germany in Kulmbach (see Fig. 1a). The water levels were measured using a levelling instrument *Ni 2* (Faig & Kahmen 2012). Based on the locations, the sites are categorized in four groups: sites 1, 2, and 3 at the river White Main; site 4 at Dobrach canal in the north; site 5 at a side canal; and sites 6, 7, and 8 at Mühl canal.

**3.3 2D HD model**

HEC-RAS 2D was used as the 2D hydrodynamic model to quantify uncertainties in the inundation. The model uses an implicit 30 finite difference solution algorithm to discretise time derivatives and hybrid approximations, combining finite differences and

finite volumes to discretise spatial derivatives (Brunner, 2010). Table 2 shows the model properties and information of the cell size. We have used the unsteady diffusive wave model presented in previous work in Bhola et al. (2018a and 2018b).

Measured discharge hydrographs described in the previous section were used as the upstream boundary condition at river gauges Ködnitz and Kauerndorf, and an energy slope value of 0.0096, based on the river slope, at the downstream boundary where the water flows out of the model domain. Along with the major rivers, canals were also represented as discharge hydrograph type.

Digital elevation model for this study was provided by the Water Management Authority, Hof and presented in Fig. 1b. In the provided elevation model, the terrain is determined by airborne laser scanning and airborne photogrammetry with a high-resolution of 1 meter, whereas the river bed was mostly recorded by the terrestrial survey. The combined elevation data were used to generate a triangulated irregular network (TIN) of the topography, which was then resampled to an irregular mesh of the 2D HD model. Special attention was given in resampling in order to preserve important features, such as rivers, dykes, buildings and roads.

**4 Results and discussion**

For the study, we have performed 1000 simulations based on uniformly distributed parameter sets for five land use classes. The sample size does contain enough samples of different behavioural models and the estimate was based on the recommendation in the literature (Aronica et al., 1998; Romanowicz and Beven, 2003) as well as the computational resources available. The HD models were simulated starting at 13.01.2011 00:00 to 14.01.2011 18:00, which requires approximately five hours to simulate an event of 42 hours on an eight-core, Intel®, Core™ 2 Duo CPU T7700 @ 2.40 cloud computer with 64 GB RAM. Eight cloud computers using the LRZ Compute Cloud, provided by the Leibniz Supercomputing Centre of the Bavarian Academy of Sciences and Humanities, were used to complete 1000 simulation in two weeks. Measured water levels at eight sites (see section 3.2) were used for the analysis of the model output. The absolute error between the simulated and measured water level is used as the goodness-of-fit to reach the objective.

**4.1 Roughness range and distribution**

The model parameter consists of roughness coefficient Manning's n for five land use classes. A simple model structure, such as diffusive wave approximation, does not represent the accurate values of roughness as this parameter is scale-dependent effective values that compensate for varying conceptual errors in the model (Néelz et al., 2009). Hence, it is recommended to use extreme feasible upper and lower ranges for the parameters in the literature (Aronica et al., 1998; Bhola et al., 2018b). In this study, ranges of Manning's n were set as: 0.015 – 0.15 for water bodies, which covers a range from very weedy reaches to rough asphalt; 0.025 – 0.110 for agriculture, short grass to medium-dense brush; 0.110 – 0.200 for forests, dense trees (Chow, 1959); 0.012 – 0.020 for transportation, firm soil to concrete; and 0.040 – 0.080, parks to gravels in urban areas

(Arcement and Schneider, 1989). Latin hypercube sampling was used to generate 1000 parameter sets using the upper and lower ranges of Manning's n set as prior and HEC-RAS 2D model was simulated for each set.

**4.2 Error tolerance**

For the analyses, the absolute error between the simulated and the measured water levels was calculated at eight sites. The simulations that produced an absolute error below a threshold at all the sites were selected. Fig. 4 shows that as we increase the threshold, the number of accepted simulations increases. To find one calibrated parameter set, the least value of tolerance can be set at 0.20 m that gives two simulations that result in the least error at all site. Having said that, the calibrated roughness set will probably hold true only for the January 2011 event as discussed in the study (Romanowicz and Beven, 2003). In order to generalize the results to other events and collect enough samples to produce uncertainty bounds, the tolerance needs to be increased. In this study, we have used 1.5 m, 0.70 and 0.50 m as the tolerance at sites to evaluate the roughness sensitivity, which results in 1000, 339 and 143 selected simulations, respectively. Nevertheless, tolerance can be changed depending on the requirements of the user. To summarize, three thresholds are used to evaluate the performance of the method in order to reduce the uncertainty bounds and are termed as follows

- Case I: Absolute error of 1.5 m resulting in 1000 simulations
- Case II: Absolute error of 0.7 m resulting in 339 simulations
- Case III: Absolute error of 0.5 m resulting in 143 simulations

**4.3 Roughness sensitivity**

The sensitivity of the model roughness was investigated, and it was observed that the sites were only sensitive to land use of water bodies and agriculture and no sensitivity was observed with respect to urban, transportation and forest. Table 3 presents the coefficient of determination ($R^2$) between Manning's n for all the land uses and absolute error for case I. Site-specific dependency in Manning's n and sites was observed for the cases in which the value of $R^2$ are found to be above *0.18* (in italic). The main reason for the lack of sensitivity can be explained by the location of the sites since they were mainly located next to bridges upstream from water bodies or agriculture land uses. Nonetheless, there are other influencing factors, such as the inundation area, velocity, and topography that could also play a role (Werner et al., 2005b). Fig. 5 shows the maximum flood inundation map for the January 2011 flood event simulated using the optimal model parameters, which were obtained by the least absolute error of 0.20 m. The inundation upstream to the sites is mainly constrained in the water bodies and agricultural land uses, which explains the impact on sensitivity of water levels to these two land uses.

[revised manuscript text omitted]

**5 Conclusions**

We have quantified the uncertainty associated with the model parameter for the flood event of January 2011 in the city of Kulmbach, Germany. Moreover, the study provides a comprehensive review of HD model uncertainty and explores the issue of high uncertainty bounds, which hinder users to analyse uncertainties. We have provided a straightforward approach to practitioners for searching model parameter spaces for behavioural models and subsequently reduce the flood inundation uncertainty bounds. Extreme ranges of model roughness in the literature were selected and 1000 uniformly distributed models were run, which resulted in wide uncertainty bounds of up to 1.26 m (90% confidence interval). To reduce the bounds, measured water levels at eight sites were used and three cases were selected on the basis of absolute error threshold values of 1.5, 0.7 and 0.5 m, which resulted in 1000, 343 and 143 accepted simulations respectively. By constraining the roughness, the bounds were reduced to a maximum of 0.34 m. In addition, the model roughness was constrained, and the physical interpretation of the constrained roughness was discussed. The model roughness was spatially distributed based on five land uses and the model was sensitive only to water bodies and agriculture.

The method is easy to incorporate into other study areas, provided that there are measured water levels available. The uncertainty analysis presented in this study allows a better understanding of the model roughness variability in HD models. The ranges researched for Manning's n in this study can represent a good starting point (prior distribution) for other studies. Our study has shown that there are significant uncertainties in HD model roughness and should be considered in decision-making. In addition, the study highlights the importance of field surveys for reducing the uncertainty in flood inundation outputs.

[revised manuscript text omitted]